# Lipschitz-Margin Training: Scalable Certification of Perturbation Invariance for Deep Neural Networks

**Yusuke Tsuzuku**
The University of Tokyo
RIKEN
tsuzuku@ms.k.u-tokyo.ac.jp

**Issei Sato**
The University of Tokyo
RIKEN
sato@k.u-tokyo.ac.jp

**Masashi Sugiyama**
RIKEN
The University of Tokyo
sugi@k.u-tokyo.ac.jp

## Abstract

High sensitivity of neural networks against malicious perturbations on inputs causes security concerns. To take a steady step towards robust classifiers, we aim to create neural network models provably defended from perturbations. Prior certification work requires strong assumptions on network structures and massive computational costs, and thus the range of their applications was limited. From the relationship between the Lipschitz constants and prediction margins, we present a computationally efficient calculation technique to lower-bound the size of adversarial perturbations that can deceive networks, and that is widely applicable to various complicated networks. Moreover, we propose an efficient training procedure that robustifies networks and significantly improves the provably guarded areas around data points. In experimental evaluations, our method showed its ability to provide a non-trivial guarantee and enhance robustness for even large networks.

## 1 Introduction

Deep neural networks are highly vulnerable against intentionally created small perturbations on inputs [29], called adversarial perturbations, which cause serious security concerns in applications such as self-driving cars. Adversarial perturbations in object recognition systems have been intensively studied [29, 10, 6], and we mainly target the object recognition systems.

One approach to defend from adversarial perturbations is to mask gradients. Defensive distillation [23], which distills networks into themselves, is one of the most prominent methods. However, Carlini and Wagner [6] showed that we can create adversarial perturbations that deceive networks trained with defensive distillation. Input transformations and detections [32, 11] are some other defense strategies, although we can bypass them [5]. Adversarial training [10, 16, 18], which injects adversarially perturbed data into training data, is a promising approach. However, there is a risk of overfitting to attacks [16, 30]. Many other heuristics have been developed to make neural networks insensitive against small perturbations on inputs. However, recent work has repeatedly succeeded to create adversarial perturbations for networks protected with heuristics in the literature [1]. For instance, Athalye et al. [2] reported that many ICLR 2018 defense papers did not adequately protect networks soon after the announcement of their acceptance. This indicates that even protected networks can be unexpectedly vulnerable, which is a crucial problem for this specific line of research because the primary concern of these studies is security threats.

The literature indicates the difficulty of defense evaluations. Thus, our goal is to ensure the lower bounds on the size of adversarial perturbations that can deceive networks for each input. Many existing approaches, which we cover in Sec. 2, are applicable only for special-structured small networks. On the other hand, common networks used in evaluations of defense methods are wide, which makes prior methods computationally intractable and complicated, which makes some prior

methods inapplicable. This work tackled this problem, and we provide a widely applicable, yet, highly scalable method that ensures large guarded areas for a wide range of network structures.

The existence of adversarial perturbations indicates that the slope of the loss landscape around data points is large, and we aim to bound the slope. An intuitive way to measure the slope is to calculate the size of the gradient of a loss with respect to an input. However, it is known to provide a false sense of security [30, 6, 2]. Thus, we require upper-bounds of the gradients. The next candidate is to calculate a local Lipschitz constant, that is the maximum size of the gradients around each data point. Even though this can provide certification, calculating the Lipschitz constant is computationally hard. We can obtain it in only small networks or get its approximation, which cannot provide certification [12, 31, 9]. A coarser but available alternative is to calculate the global Lipschitz constant. However, prior work could provide only magnitudes of smaller certifications compared to the usual discretization of images even for small networks [29, 24]. We show that we can overcome such looseness with our improved and unified bounds and a developed training procedure. The training procedure is more general and effective than previous approaches [7, 33]. We empirically observed that the training procedure also improves robustness against current attack methods.

## 2   Related work

In this section, we review prior work to provide certifications for networks. One of the popular approaches is restricting discussion to networks using ReLU [20] exclusively as their activation functions and reducing the verification problem to some other well-studied problems. Bastani et al. [4] encoded networks to linear programs, Katz et al. [14, 13] reduced the problem to Satisfiability Modulo Theory, and Raghunathan et al. [25] encoded networks to semidefinite programs. However, these formulations demand prohibitive computational costs and their applications are limited to only small networks. As a relatively tractable method, Kolter and Wong [15] has bounded the influence of $\ell_\infty$-norm bounded perturbations using convex outer-polytopes. However, it is still hard to scale this method to deep or wide networks. Another approach is assuming smoothness of networks and losses. Hein and Andriushchenko [12] focused on local Lipschitz constants of neural networks around each input. However, the guarantee is provided only for networks with one hidden layer. Sinha et al. [27] proposed a certifiable procedure of adversarial training. However, smoothness constants, which their certification requires, are usually unavailable or infinite. As a concurrent work, Ruan et al. [26] proposed another algorithm to certify robustness with more scalable manner than previous approaches. We note that our algorithm is still significantly faster.

## 3   Problem formulation

We define the threat model, our defense goal, and basic terminologies.

**Threat model:**   Let $X$ be a data point from data distribution $D$ and its true label be $t_X \in \{1, \ldots, K\}$ where $K$ is the number of classes. Attackers create a new data point similar to $X$ which deceives defenders' classifiers. In this paper, we consider the $\ell_2$-norm as a similarity measure between data points because it is one of the most common metrics [19, 6].

Let $c$ be a positive constant and $F$ be a classifier. We assume that the output of $F$ is a vector $F(X)$ and the classifier predicts the label with $\mathrm{argmax}_{i \in \{1,\ldots,K\}} \{F(X)_i\}$, where $F(X)_i$ denotes the $i$-th element of $F(X)$. Now, we define adversarial perturbation $\epsilon_{F,X}$ as follows.

$$\epsilon_{F,X} \in \left\{ \epsilon \left| \|\epsilon\|_2 < c \ \wedge \ t_X \neq \mathrm{argmax}_{i \in \{1,\ldots,K\}} \{F(X+\epsilon)_i\} \right. \right\}.$$

**Defense goal:**   We define a guarded area for a network $F$ and a data point $X$ as a hypersphere with a radius $c$ that satisfies the following condition:

$$\forall \epsilon, \left( \|\epsilon\|_2 < c \ \Rightarrow \ t_X = \mathrm{argmax}_{i \in \{1,\ldots,K\}} \{F(X+\epsilon)_i\} \right). \tag{1}$$

This condition (1) is always satisfied when $c = 0$. Our goal is to ensure that neural networks have larger guarded areas for data points in data distribution.

# 4  Calculation and enlargement of guarded area

In this section, we first describe basic concepts for calculating the provably guarded area defined in Sec. 3. Next, we outline our training procedure to enlarge the guarded area.

## 4.1  Lipschitz constant and guarded area

We explain how to calculate the guarded area using the Lipschitz constant. If $L_F$ bounds the Lipschitz constant of neural network $F$, we have the following from the definition of the Lipschitz constant:

$$\|F(X) - F(X + \epsilon)\|_2 \leq L_F \|\epsilon\|_2.$$

Note that if the last layer of $F$ is softmax, we only need to consider the subnetwork before the softmax layer. We introduce the notion of prediction margin $M_{F,X}$:

$$M_{F,X} := F(X)_{t_X} - \max_{i \neq t_X}\{F(X)_i\}.$$

This margin has been studied in relationship to generalization bounds [17, 3, 22]. Using the prediction margin, we can prove the following proposition holds.

**Proposition 1.**

$$(M_{F,X} \geq \sqrt{2}L_F\|\epsilon\|_2) \Rightarrow (M_{F,X+\epsilon} \geq 0). \tag{2}$$

The details of the proof are in Appendix A of the supplementary material. Thus, perturbations smaller than $M_{F,X} / (\sqrt{2}L_F)$ cannot deceive the network $F$ for a data point $X$. Proposition 1 sees network $F$ as a function with a multidimensional output. This connects the Lipschitz constant of a network, which has been discussed in Szegedy et al. [29] and Cisse et al. [7], with the absence of adversarial perturbations. If we cast the problem to a set of functions with a one-dimensional output, we can obtain a variant of Prop. 1. Assume that the last layer before softmax in $F$ is a fully-connected layer and $w_i$ is the $i$-th raw of its weight matrix. Let $L_{\text{sub}}$ be a Lipschitz constant of a sub-network of $F$ before the last fully-connected layer. We obtain the following proposition directly from the definition of the Lipschitz constant [12, 31].

**Proposition 2.**

$$(\forall i, (F_{t_X} - F_i \geq L_{\text{sub}}\|w_{t_X} - w_i\|_2\|\epsilon\|_2)) \Rightarrow (M_{F,X+\epsilon} \geq 0). \tag{3}$$

We can use either Prop. 1 or Prop. 2 for the certification. Calculations of the Lipschitz constant, which is not straightforward in large and complex networks, will be explained in Sec. 5.

## 4.2  Guarded area enlargement

To ensure non-trivial guarded areas, we propose a training procedure that enlarges the provably guarded area.

**Lipschitz-margin training:**  To encourage conditions Eq.(2) or Eq.(3) to be satisfied with the training data, we convert them into losses. We take Eq.(2) as an example. To make Eq.(2) satisfied for perturbations with $\ell_2$-norm larger than $c$, we require the following condition.

$$\forall i \neq t_X, (F_{t_X} \geq F_i + \sqrt{2}cL_F). \tag{4}$$

Thus, we add $\sqrt{2}cL_F$ to all elements in logits except for the index corresponding to $t_X$. In training, we calculate an estimation of the upper bound of $L_F$ with a computationally efficient and differentiable way and use it instead of $L_F$. Hyperparameter $c$ is specified by users. We call this training procedure *Lipschitz-margin training* (LMT). The algorithm is provided in Figure 1. Using Eq.(3) instead of Eq.(2) is straightforward. Small additional techniques to make LMT more stable is given in Appendix E of the supplementary material.

**Interpretation of LMT:**  From the former paragraph, we can see that LMT maximizes the number of training data points that have larger guarded areas than $c$, as long as the original training procedure maximizes the number of them that are correctly classified. We experimentally evaluate its generalization to test data in Sec. 6. The hyperparameter $c$ is easy to interpret and easy to tune. The larger

$c$ we specify, the stronger invariant property the trained network will have. However, this does not mean that the trained network always has high accuracy against noisy examples. To see this, consider the case where $c$ is extremely large. In such a case, constant functions become an optimal solution. We can interpret LMT as an interpolation between the original function, which is highly expressive but extremely non-smooth, and constant functions, which are robust and smooth.

**Computational costs:**   A main computational overhead of LMT is the calculation of the Lipschitz constant. We show in Sec. 5 that its computational cost is almost the same as increasing the batch size by one. Since we typically have tens or hundreds of samples in a mini-batch, this cost is negligible.

## 5   Calculation of the Lipschitz constant

In this section, we first describe a method to calculate upper bounds of the Lipschitz constant. We bound the Lipschitz constant of each component and recursively calculate the overall bound. The concept is from Szegedy et al. [29]. While prior work required separate analysis for slightly different components [29, 24, 7, 26], we provide a more unified analysis. Furthermore, we provide a fast calculation algorithm for both the upper bounds and their differentiable approximation.

### 5.1   Composition, addition, and concatenation

We describe the relationships between the Lipschitz constants and some functionals which frequently appears in deep neural networks: composition, addition, and concatenation. Let $f$ and $g$ be functions with Lipschitz constants bounded by $L_1$ and $L_2$, respectively. The Lipschitz constant of output for each functional is bounded as follows:

$$\text{composition}\quad f \circ g : L_1 \cdot L_2, \quad \text{addition}\quad f + g\quad : L_1 + L_2, \quad \text{concatenation}\quad (f, g) : \sqrt{L_1^2 + L_2^2}.$$

### 5.2   Major Components

We describe bounds of the Lipschitz constants of major layers commonly used in image recognition tasks. We note that we can ignore any bias parameters because they do not change the Lipschitz constants of each layer.

**Linear layers in general:**   Fully-connected, convolutional and normalization layers are typically linear operations at inference time. For instance, batch-normalization is a multiplication of a diagonal matrix whose $i$-th element is $\gamma_i / \sqrt{\sigma_i^2 + \epsilon}$, where $\gamma_i, \sigma_i^2, \epsilon$ are a scaling parameter, running average of variance, and a constant, respectively. Since the composition of linear operators is also linear, we can jointly calculate the Lipschitz constant of some common pairs of layers such as convolution + batch-normalization. By using the following theorem, we proposed a more unified algorithm than Yoshida and Miyato [33].

**Theorem 1.** *Let $\phi$ be a linear operator from $\mathbb{R}^n$ to $\mathbb{R}^m$, where $n < \infty$ and $m < \infty$. We initialize a vector $u \in \mathbb{R}^n$ from a Gaussian with zero mean and unit variance. When we iteratively apply the*

---

**Algorithm 1:** Lipschitz-margin training

**hyperparam :** $c$ : required robustness
**input**        **:** $X$ : image, $t_X$: label of $X$
$y \leftarrow \text{Forward}(X)$;
$L \leftarrow \text{CalcLipschitzConst}()$;
**foreach** *index $i$***:**
  **if** $i \neq t_X$**:**
    $y_i \mathrel{+}= \sqrt{2}Lc$;
$p \leftarrow \text{SoftmaxIfNecessary}(y)$;
$\ell \leftarrow \text{CalcLoss}(p, t_X)$;

Figure 1:  Lipschitz-margin training algorithm when we use Prop. 1.

---

**Algorithm 2:** Calculation of operator norm

**input**        **:** $u$ : array at previous
                    iteration
**target**       **:** $f$ : linear function
$u \leftarrow u/\|u\|_2$;
// $\sigma$ is an approximated spectral norm;
$\sigma \leftarrow \|f(u)\|_2$;
$L \leftarrow \text{CalcLipschitzConst}(\sigma)$;
$\ell \leftarrow \text{CalcLoss}(L)$;
$u \leftarrow \frac{\partial \ell}{\partial u}$;

Figure 2:  Calculation of the spectral norm of linear components at training time.

*following update formula, the $\ell_2$-norm of $u$ converges to the square of the operator norm of $\phi$ in terms of $\ell_2$-norm, almost surely.*

$$u \leftarrow u/\|u\|_2, \quad v \leftarrow \phi(u), \quad u \leftarrow \frac{1}{2}\frac{\partial \|v\|_2^2}{\partial u}.$$

The proof is found in Appendix C.1 of the supplementary material. The algorithm for training time is provided in Figure 2. At training time, we need only one iteration of the above update formula as with Yoshida and Miyato [33]. Note that for estimation of the operator norm for a forward path, we do not require to use gradients. In a convolutional layer, for instance, we do not require another convolution operation or transposed convolution. We only need to increase the batch size by one. The wide availability of our calculation method will be especially useful when more complicated linear operators than usual convolution appear in the future. Since we want to ensure that the calculated bound is an *upper-bound* for certification, we can use the following theorem.

**Theorem 2.** *Let $\|\phi\|_2$ and $R_k$ be an operator norm of a function $\phi$ in terms of the $\ell_2$-norm and the $\ell_2$-norm of the vector $u$ at the $k$-th iteration, where each element $u$ is initialized by a Gaussian with zero mean and unit variance. With probability higher than $1 - \sqrt{2/\pi}$, the error between $\|\phi\|_2^2$ and $R_k$ is smaller than $(\Delta_k + \sqrt{\Delta_k(4R_k + \Delta_k)})/2$, where $\Delta_k := (R_k - R_{k-1})n$.*

The proof is in Appendix C.3 of the supplementary material, which is mostly from Friedman [8]. If we use a large batch for the power iteration, the probability becomes exponentially closer to one. We can also use singular value decomposition as another way for accurate calculation. Despite its simplicity, the obtained bound for convolutional layers is much tighter than the previous results in Peck et al. [24] and Cisse et al. [7], and that for normalization layers is novel. We numerically confirm the improvement of bounds in Sec. 6.

**Pooling and activation:**   First, we have the following theorem.

**Theorem 3.** *Define $f(Z) = (f_1(Z^1), f_2(Z^2), ..., f_\Lambda(Z^\Lambda))$, where $Z^\lambda \subset Z$ and $\|f_\lambda\|_2 \leq L$ for all $\lambda$. Then,*

$$\|f\|_2 \leq \sqrt{n}L,$$

*where $n := \max_j |\{\lambda | x_j \in Z^\lambda\}|$ and $x_j$ is the $j$-th element of $x$.*

The proof, whose idea comes from Cisse et al. [7], is found in Appendix D.1 of the supplementary material. The exact form of $n$ in the pooling and convolutional layers is given in Appendix D.3 of the supplementary material. The assumption in Theorem 3 holds for most layers of networks for image recognition tasks, including pooling layers, convolutional layers, and activation functions. Careful counting of $n$ leads to improved bounds on the relationship between the Lipschitz constant of a convolutional layer and the spectral norm of its reshaped kernel from the previous result [7].

**Corollary 1.** *Let $\|\mathrm{Conv}\|_2$ be the operator norm of a convolutional layer in terms of the $\ell_2$-norm, and $\|W'\|_2$ be the spectral norm of a matrix where the kernel of the convolution is reshaped into a matrix with the same number of rows as its output channel size. Assume that the width and the height of its input before padding are larger or equal to those of the kernel. The following inequality holds.*

$$\|W'\|_2 \leq \|\mathrm{Conv}\|_2 \leq \sqrt{n}\|W'\|_2,$$

*where $n$ is a constant independent of the weight matrix.*

The proof of Corollary 1 is in Appendix D.2 of the supplementary material. Lists of the Lipschitz constant of pooling layers and activation functions are summarized in Appendix D of the supplementary material.

### 5.3  Putting them together

With recursive computation using the bounds described in the previous sections, we can calculate an upper bound of the Lipschitz constants of the whole network in a differentiable manner with respect to network parameters. At inference time, calculation of the Lipschitz constant is required only once.

In calculations at training time, there may be some notable differences in the Lipschitz constants. For example, $\sigma_i$ in a batch normalization layer depends on its input. However, we empirically found that calculating the Lipschitz constants using the same bound as inference time effectively regularizes the Lipschitz constant. This lets us deal with batch-normalization layers, which prior work ignored despite its impact on the Lipschitz constant [7, 33].

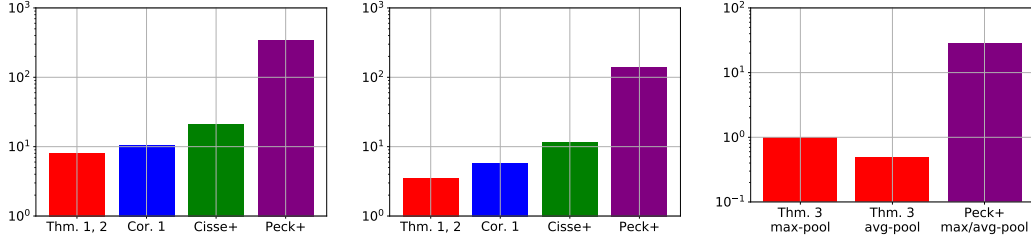

Figure 3: Comparison of bounds in layers. Left: the second convolutional layer of a naive model in Sec 6. Center: the second convolutional layer of an LMT model in Sec 6. Right: pooling layers assuming size 2 and its input size is $28 \times 28$.

## 6 Numerical evaluations

In this section, we show the results of numerical evaluations. Since our goal is to create networks with stronger certification, we evaluated the following three points.

1. Our bounds of the Lipschitz constants are tighter than previous ones (Sec. 6.1).
2. LMT effectively enlarges the provably guarded area (Secs. 6.1 and 6.2).
3. Our calculation technique of the guarded area and LMT are available for modern large and complex networks (Sec. 6.2).

We also evaluated the robustness of trained networks against current attacks and confirmed that LMT robustifies networks (Secs. 6.1 and 6.2). For calculating the Lipschitz constant and guarded area, we used Prop. 2. Detailed experimental setups are available in Appendix F of the supplementary material. Our codes are available at https://github.com/ytsmiling/lmt.

### 6.1 Tightness of bounds

We numerically validated improvements of bounds for each component and numerically analyzed the tightness of overall bounds of the Lipschitz constant. We also see the non-triviality of the provably guarded area. We used the same network and hyperparameters as Kolter and Wong [15].

**Improvement in each component:** We evaluated the difference of bounds in convolutional layers in networks trained using a usual training procedure and LMT. Figure 3 shows comparisons between the bounds in the second convolutional layer. It also shows the difference of bounds in pooling layers, which does not depend on training methods. We can confirm improvement in each bound. This results in significant differences in upper-bounds of the Lipschitz constants of the whole networks.

**Analysis of tightness:** Let $L$ be an upper-bound of the Lipschitz constant calculated by our method. Let $L_{\text{local}}, L_{\text{global}}$ be the local and global Lipschitz constants. Between them, we have the following relationship.

$$\underbrace{\frac{\text{Margin}}{L}}_{(A)} \underset{(i)}{\leq} \underbrace{\frac{\text{Margin}}{L_{\text{global}}}}_{(B)} \underset{(ii)}{\leq} \underbrace{\frac{\text{Margin}}{L_{\text{local}}}}_{(C)} \underset{(iii)}{\leq} \underbrace{\text{Smallest Adversarial Perturbation}}_{(D)} \tag{5}$$

We analyzed errors in inequalities (i) – (iii). We define an error of (i) as (B)/(A) and others in the same way. We used lower bounds of the local and global Lipschitz constant calculated by the maximum size of gradients found. A detailed procedure for the calculation is explained in Appendix F.1.3 of the supplementary material. For the generation of adversarial perturbations, we used DeepFool [19]. Note that (iii) does not hold because we calculated mere lower bounds of Lipschitz constants in (B) and (C). We analyzed inequality (5) in an unregularized model, an adversarially trained (AT) model with the 30-iteration C&W attack [6], and an LMT model. Figure 4 shows the result. With an unregularized model, estimated error ratios in (i) – (iii) were 39.9, 1.13, and 1.82 respectively. This shows that even if we could precisely calculate the local Lipschitz constant for each data point with

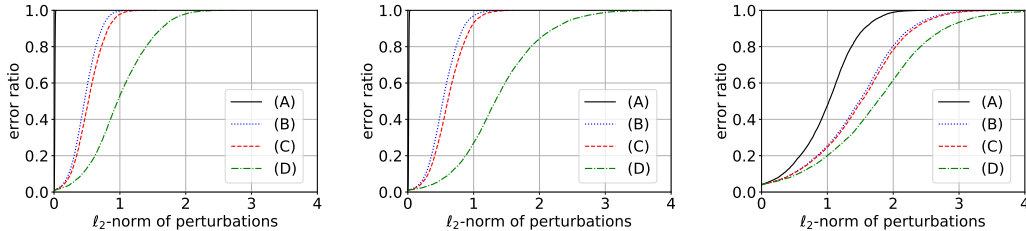

Figure 4: Comparison of error bounds using inequalities (5) with estimation. Each label corresponds to the value in inequality (5). Left: naive model, Center: AT model, Right: LMT model.

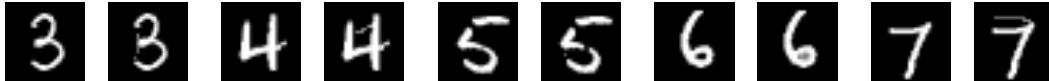

Figure 5: Examples of pairs of an original image and an artificially perturbed image which LMT model was ensured not to make wrong predictions. The differences between images are large and visually perceptible. On the basis of Proposition 2, any patterns of perturbations with the same or smaller magnitudes could not deceive the network trained with LMT.

possibly substantial computational costs, inequality (iii) becomes more than $1.8$ times looser than the size of adversarial perturbations found by DeepFool. In an AT model, the discrepancy became more than $2.4$. On the other hand, in an LMT model, estimated error ratios in (i) – (iii) were $1.42$, $1.02$, and $1.15$ respectively. The overall median error between the size of found adversarial perturbations, and the provably guarded area was $1.72$. This shows that the trained network became smooth and Lipschitz constant based certifications became significantly tighter when we use LMT. This also resulted in better defense against attack. For reference, the median of found adversarial perturbations for an unregularized model was $0.97$, while the median of the size of the provably guarded area was $1.02$ in an LMT model.

**Size of provably guarded area:** We discuss the size of the provably guarded area, which is practically more interesting than tightness. While our algorithm has clear advantages on computational costs and broad applicability over prior work, guarded areas that our algorithm ensured were non-trivially large. In a naive model, the median of the size of perturbations we could certify invariance was $0.012$. This means changing several pixels by one in usual $0$–$255$ scale cannot change their prediction. Even though this result is not so tight as seen in the previous paragraph, this is significantly larger than prior computationally cheap algorithm proposed by Peck et al. [24]. The more impressive result was obtained in models trained with LMT, and the median of the guarded area was $1.02$. This corresponds to $0.036$ in the $\ell_\infty$ norm. Kolter and Wong [15], which used the same network and hyperparameters as ours, reported that they could defend from perturbations with its $\ell_\infty$-norm bounded by $0.1$ for more than $94\%$ examples. Thus, in the $\ell_\infty$-norm, our work is inferior, if we ignore their limited applicability and massive computational demands. However, our algorithm mainly targets the $\ell_2$-norm, and in that sense, the guarded area is significantly larger. Moreover, for more than half of the test data, we could ensure that there are no one-pixel attacks [28]. To confirm the non-triviality of the obtained certification, we have some examples of provably guarded images in Figure 5.

## 6.2 Scalability test

We evaluated our method with a larger and more complex network to confirm its broad applicability and scalability. We used 16-layered wide residual networks [34] with width factor $4$ on the SVHN dataset [21] following Cisse et al. [7]. To the best of our knowledge, this is the largest network concerned with certification. We compared LMT with a naive counterpart, which uses weight decay, spectral norm regularization [33], and Parseval networks.

**Size of provably guarded area:** For a model trained with LMT, we could ensure larger guarded areas than $0.029$ for more than half of test data. This order of certification was only provided for

Table 1: Accuracy of trained wide residual networks on SVHN against C&W attack.

| | Clean | Size of perturbations | | |
| | | 0.2 | 0.5 | 1.0 |
|---|---|---|---|---|
| weight decay | **98.31** | 72.38 | 20.98 | 2.02 |
| Parseval network | 98.30 | 71.35 | 17.92 | 0.94 |
| spectral norm regularization | 98.27 | 73.66 | 20.35 | 1.39 |
| **LMT** | 96.38 | **86.90** | **55.11** | **17.69** |

small networks in prior work. In models trained with other methods, we could not provide such strong certification. There are mainly two differences between LMT and other methods. First, LMT enlarges prediction margins. Second, LMT regularizes batch-normalization layers, while in other methods, batch-normalization layers cancel the regularization on weight matrices and kernel of convolutional layers. We also conducted additional experiments to provide further certification for the network. First, we replaced convolution with kernel size 1 and stride 2 with average-pooling with size 2 and convolution with kernel size 1. Then, we used LMT with $c = 0.1$. As a result, while the accuracy dropped to $86\%$, the median size of the provably guarded areas was larger than 0.08. This corresponds to that changing $400$ elements of input by $\pm 1$ in usual image scales (0–255) cannot cause error over $50\%$ for the trained network. These certifications are non-trivial, and to the best of our knowledge, these are the best certification provided for this large network.

**Robustness against attack:** We evaluated the robustness of trained networks against adversarial perturbations created by the current attacks. We used C&W attack [6] with 100 iterations and no random restart for evaluation. Table 1 summarizes the results. While LMT slightly dropped its accuracy, it largely improved robustness compared to other regularization based techniques. Since these techniques are independent of other techniques such as adversarial training or input transformations, further robustness will be expected when LMT is combined with them.

# 7 Conclusion

To ensure perturbation invariance of a broad range of networks with a computationally efficient procedure, we achieved the following.

1. We offered general and tighter spectral bounds for each component of neural networks.
2. We introduced general and fast calculation algorithm for the upper bound of operator norms and its differentiable approximation.
3. We proposed a training algorithm which effectively constrains networks to be smooth, and achieves better certification and robustness against attacks.
4. We successfully provided non-trivial certification for small to large networks with negligible computational costs.

We believe that this work will serve as an essential step towards both certifiable and robust deep learning models. Applying developed techniques to other Lipschitz-concerned domains such as training of GAN or training with noisy labels is future work.

## Acknowledgement

Authors appreciate Takeru Miyato for valuable feedback. YT was supported by Toyota/Dwango AI scholarship. IS was supported by KAKENHI 17H04693. MS was supported by KAKENHI 17H00757.

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
