[Supplementary Material]

# Appendix of Lipschitz-Margin Training

## Contents

# A   Proof of Proposition 1

We prove Prop. 1 in Sec. 4.1. Let us consider a classifier with Lipschitz constant $L$. Let $F(X)$ be an output vector of the classifier for a data point $X$.

The statement to prove is the following:

$$F(X)_{t_X} - \max_{i \neq t_X} F(X)_i \geq \sqrt{2}L\|\epsilon\|_2 \Rightarrow F(X+\epsilon)_{t_X} - \max_{i \neq t_X} F(X+\epsilon)_i \geq 0. \tag{1}$$

If we prove the following, it suffices:

$$F(X+\epsilon)_{t_X} - \max_{i \neq t_X} F(X+\epsilon)_i \geq F(X)_{t_X} - \max_{i \neq t_X} F(X)_i - \sqrt{2}L\|\epsilon\|_2. \tag{2}$$

Before proving inequality (2), we have the following lemma.

**Lemma 1.** *For real vectors $x$ and $y$, the following inequality holds:*

$$\left|\max_{i \neq t_X} x_i - \max_{i \neq t_X} y_i\right| \leq \max_{i \neq t_X}|x_i - y_i|.$$

*Proof.* W.l.o.g. we assume $\max_{i \neq t_X} x_i \geq \max_{i \neq t_X} y_i$. Let $j$ be $\operatorname{argmax}_{i \neq t_X} x_i$. Then,

$$\begin{aligned}
\left|\max_{i \neq t_X} x_i - \max_{i \neq t_X} y_i\right| &= \max_{i \neq t_X} x_i - \max_{i \neq t_X} y_i \\
&= x_j - \max_{i \neq t_X} y_i \\
&\leq x_j - y_j \\
&\leq \max_{i \neq t_X}|x_i - y_i|
\end{aligned}$$

$\square$

Now, we can prove the inequality (2).

$$\begin{aligned}
&F(X+\epsilon)_{t_X} - \max_{i \neq t_X} F(X+\epsilon)_i \\
=&F(X)_{t_X} - \max_{i \neq t_X} F(X)_i + (F(X+\epsilon)_{t_X} - F(X)_{t_X}) - \left(\max_{i \neq t_X} F(X+\epsilon)_i - \max_{i \neq t_X} F(X)_i\right) \\
\geq&F(X)_{t_X} - \max_{i \neq t_X} F(X)_i - |F(X+\epsilon)_{t_X} - F(X)_{t_X}| - |\max_{i \neq t_X} F(X+\epsilon)_i - \max_{i \neq t_X} F(X)_i| \\
\geq&F(X)_{t_X} - \max_{i \neq t_X} F(X)_i - |F(X+\epsilon)_{t_X} - F(X)_{t_X}| - \max_{i \neq t_X}|F(X+\epsilon)_i - F(X)_i| \\
\geq&F(X)_{t_X} - \max_{i \neq t_X} F(X)_i - \max_{a_1, a_2 \in \mathbb{R}}\left\{|a_1| + |a_2| \,\middle|\, \sqrt{a_1^2 + a_2^2} \leq L\|\epsilon\|_2\right\} \\
=&F(X)_{t_X} - \max_{i \neq t_X} F(X)_i - \sqrt{2}L\|\epsilon\|_2.
\end{aligned}$$

$\square$

# B   Lipschitz constant of basic functionals

We prove bounds described in Sec. 5.1. Let $f$ and $g$ be functions with their Lipschitz constants bounded with $L_1$ and $L_2$, respectively.

### B.1 Composition of functions

$$
\frac{\|f(g(X_1)) - f(g(X_2))\|_2}{\|X_1 - X_0\|_2} = \frac{\|f(g(X_1)) - f(g(X_2))\|_2}{\|g(X_1) - g(X_2)\|_2} \cdot \frac{\|g(X_1) - g(X_2)\|_2}{\|X_1 - X_0\|_2}
$$
$$
\leq L_1 \cdot L_2.
$$

### B.2 Addition of functions

Using triangle inequality,
$$
\frac{\|(f+g)(X_1) - (f+g)(X_2))\|_2}{\|X_1 - X_0\|_2} \leq \frac{\|f(X_1) - f(X_2)\|_2 + \|g(X_1) - g(X_2)\|_2}{\|X_1 - X_0\|_2}
$$
$$
\leq L_1 + L_2.
$$

### B.3 Concatenation of functions

$$
\frac{\|(f(X_1), g(X_1)) - (f(X_0), g(X_0))\|_2}{\|X_1 - X_0\|_2} = \sqrt{\frac{\|(f(X_1), g(X_1)) - (f(X_0), g(X_0))\|_2^2}{\|X_1 - X_0\|_2^2}}
$$
$$
= \sqrt{\frac{\|f(X_1) - f(X_0)\|_2^2 + \|g(X_1) - g(X_0)\|_2^2}{\|X_1 - X_0\|_2^2}}
$$
$$
\leq \sqrt{L_1^2 + L_2^2}.
$$

## C  Lipschitz constant of linear components

We see the Lipschitz constant of linear components, given in Sec. 5.2, in more detail. We first prove Theorem 1, and Theorem 2. Next, we focus on its calculation for normalization layers.

### C.1  Proof of Theorem 1

Since there exists a matrix representation $M$ of $\phi$ and the operator norm of $\phi$ in terms of $\ell_2$-norm is equivalent to the spectral norm of $M$, considering $v = Mu$ is sufficient. Now, we have
$$
\frac{1}{2}\frac{\partial \|v\|_2^2}{\partial u} = \frac{1}{2}\frac{\partial (u^\top M^\top M u)}{\partial u}
$$
$$
= M^\top M u.
$$
Thus, recursive application of the algorithm in Theorem 1 is equivalent to the power iteration to $M^\top M$. Since the maximum eigen value of $M^\top M$ is a square of the spectral norm of $M$, $u$ converges to the square of the spectral norm of $M$ almost surely in the algorithm. $\qquad\square$

### C.2  Explanation of Algorithm 2

We use the same notation with Algorithm 2. In Algorithm 2, we only care the direction of the vector $u$ because we normalize it at every iteration. We first explain that the direction of $u$ converges to a singular vector of the largest singular value of the linear function $f$ when $f$ is fixed.
Since
$$
\frac{\partial L}{\partial u} = \frac{\partial L}{\partial \sigma} \cdot \frac{\partial \sigma}{\partial u}
$$
$$
= 2\frac{\partial L}{\partial \sigma} \cdot \frac{\partial \sigma}{\partial \sigma^2} \cdot \left(\frac{1}{2}\frac{\partial \sigma^2}{\partial u}\right)
$$
and $2\frac{\partial L}{\partial \sigma} \cdot \frac{\partial \sigma}{\partial \sigma^2}$ is a scalar, $u$ converges to the same direction with Theorem 1. In other words, $u$ converges to the singular vector of the largest singular of $M$.

If $u$ approximates the singular vector, then $\sigma := \|f(u)\|_2$ approximates the spectral norm of $f$. If $f$ changes a little per iteration, even though Algorithm 2 performs only one step of the power iteration per iteration, we can keep good approximation of the spectral norm [13].

## C.3 Proof of Theorem 2

From the proof of Theorem 1 in Appendix C.1, we considers power iteration to $M^\top M$. Let $\lambda_1$ be the largest singular value of a matrix $M^\top M$. Since $M^\top M$ is a symmetric positive definite matrix, from Theorem 1.1 in Friedman [4], we have

$$\lambda_1 - R_k \leq \frac{1}{2}\left(\Delta_k + \sqrt{\Delta_k\left(4R_k + \Delta_k\right)}\right),$$

where $\Delta_k$ is bounded by $\omega - 1$ from Prop. 2.2 in [4]. A quantity $\omega$ has the following relationship [4]:

$$\Pr(\omega - 1 \geq n) \leq \sqrt{2/\pi}.$$

Thus, the Theorem 2 holds. $\qquad\square$

If we use batchsize $128$ for the algorithm and take the max of all upper bound, then the failure probability is less than $(2/\pi)^{128/2} \leq 10^{-12}$.

## C.4 Calculation of normalization layers

### C.4.1 Example: batch-normalization

Batch normalization applies the following function,

$$x_i \leftarrow \gamma_i \frac{x_i - \mu_i}{\sqrt{\sigma_i^2 + \epsilon}} + \beta_i, \tag{3}$$

where $\gamma_i$ and $\beta_i$ are learnable parameters and $\mu_i, \sigma_i$ are the mean and deviation of (mini) batch, respectively. Parameters and variables $\gamma_i, \beta_i, \mu_i$, and $\sigma_i$ are constant at the inference time. Small constant $\epsilon$ is generally added for numerical stability. We can rewrite an update of (3) as follows:

$$x_i \leftarrow \frac{\gamma_i}{\sqrt{\sigma_i^2 + \epsilon}}x_i + \left(-\gamma_i\frac{\mu_i}{\sqrt{\sigma_i^2 + \epsilon}} + \beta_i\right).$$

Since the second term is constant in terms of input, it is independent of the Lipschitz constant. Thus, we consider the following update:

$$x_i \leftarrow \frac{\gamma_i}{\sqrt{\sigma_i^2 + \epsilon}}x_i.$$

The Lipschitz constant can be bounded by $\max_i\{|\gamma_i|/\sqrt{\sigma_i^2 + \epsilon}\}$.

Since the opertion is linear, we can also use Algorithm 2 for the calculation. This allows us to calculate the Lipschitz constant of batch-noramlization and precedent other linear layers jointly. When we apply the algorithm 2 to a single batch-normalization layer, a numerical issue can offer. See Appendix C.4.3 for more details.

### C.4.2 Other normalizations

In weight normalization [11], the same discussion applies if we replace $\sqrt{\sigma_i^2 + \epsilon}$ in batch-normalization with $\|w_i\|_2$, where $w_i$ is the $i$-th row of a weight matrix.

### C.4.3 Undesired convergence of power iteration

In some cases, estimation of spectral norm using power iteration can fail in training dynamics. For example, in batch-normalization layer, $u$ in Algorithm 2 converges to some one-hot vector. Once $u$ converges, no matter how much other parameters change during training, $u$ stay the same. To avoid the problem, when we apply Algorithm 2 to normalization layers, we added small perturbations on a vector $u$ in the algorithm at every iteration after its normalization.

# D Lipschitz constant of pooling and activation

## D.1 Proof of Theorem 3

First, we prove the following lemma:

**Lemma 2.** *Let vector $X$ be a concatenation of vectors $X_\lambda (0 \le \lambda \le n)$ and let $f$ be a function such that $f(x)$ is a concatenations of vectors $f(X_\lambda)$, where each $f_\lambda$ is a function with its Lipschitz constant bounded by $L$. Then, the Lipschitz constant of $f$ can be bounded by $L$.*

$$\frac{\|f(X) - f(Y)\|_2^2}{\|X - Y\|_2^2} = \frac{\sum_\lambda \|f_\lambda(X_\lambda) - f_\lambda(Y_\lambda)\|_2^2}{\|X - Y\|_2^2}$$

$$\le \frac{\sum_\lambda L^2 \|X_\lambda - Y_\lambda\|_2^2}{\|X - Y\|_2^2}$$

$$= L^2$$

$\square$

Since n-th time repetition is the same with n-th time concatenation, which is explained in Appendix B.3, its Lipschitz constant is bounded by $\sqrt{n}$. Using Appendix B.3 and the above lemma, we obtain the bound in Theorem 3. $\square$

### D.2 Proof of Corollary 1

**notation:** $ch_{in}, ch_{out}, h_k, w_k$: input channel size, output channel size, kernel height, kernel width. $W'$: a matrix which kernel of a convolution $W \in R^{ch_{out} \times ch_{in} \times h_k \times w_k}$ is reshaped into the size $ch_{out} \times (ch_{in} \times h_k \times w_k)$.

**proof:** The operation in a convolution layer satisfies the assumption in Theorem 3, where all $f_i$ are the matrix multiplication of $W'$. Thus, the right inequality holds. Since matrix multiplication with $W'$ is applied at least once in the convolution, the left inequality holds. $\square$

This result is similar to Cisse et al. [2], but we can provide better bounds by carefully calculating the number of repetition, given in Appendix D.3.

### D.3 Tighter bound of n-repetition in Theorem 3

We provide tight number of the repetition for pooling and convolutional layers here.

**notation:** $h_{in}, w_{in}$: height and width of input array.
$h_k, w_k$: kernel height, kernel width.
$h_s, w_s$: stride height, stride width.

**number of repetition:**

$$\left\lceil \frac{\min(h_k, h_{in} - h_k + 1)}{h_s} \right\rceil \cdot \left\lceil \frac{\min(w_k, w_{in} - w_k + 1)}{w_s} \right\rceil.$$

**derivation:** First of all, the repetition is bounded by the size of reception field, which is $h_k \cdot w_k$. This is provided by Cisse et al. [2]. Now, we extend the bound by considering the input size and stride. Firstly, we consider the input size after padding. If both the input and kernel size are $8 \times 8$, the number of repetition is obviously bounded by $1$. Similarly, the number of repetition can be bounded by the following:

$$\min(h_k, h_{in} - h_k + 1) \cdot \min(w_k, w_{in} - w_k + 1).$$

We can further bound the time of repetition by considering the stride as follows:

$$\left\lceil \frac{\min(h_k, h_{in} - h_k + 1)}{h_s} \right\rceil \cdot \left\lceil \frac{\min(w_k, w_{in} - w_k + 1)}{w_s} \right\rceil.$$

$\square$

Table 1: Lipschitz constants of major activation functions.

| Activation | Lipschitz constant |
|---|---|
| ReLU | 1 |
| Leaky ReLU [8] | $\max(1, \lvert\alpha\rvert)$ |
| sigmoid | $1/4$ |
| tanh | 1 |
| soft plus [1] | 1 |
| ELU [3] | $\max(1, \lvert\alpha\rvert)$ |

### D.3.1 The Lipschitz constant of $f_i$ in Theorem 3 for Pooling layers

**max-pooling:** Lipschitz constant of max function is bounded by one.

**average-pooling:** Before bounding the Lipschitz constant, we note that the following inequality holds for a vector $X$:

$$\left(\sum_{i=1}^{n} X_i\right)^2 \leq n \sum_{i=1}^{n} X_i^2.$$

This can be proved using

$$\frac{1}{n}\sum_{i=1}^{n} X_i^2 - \left(\frac{1}{n}\sum_{i=1}^{n} X_i\right)^2 = \frac{1}{n}\sum_{i=1}^{n}\left(X_i - \frac{1}{n}\sum_{i=1}^{n} X_i\right)^2 \geq 0.$$

Now, we bound the Lipschitz constant of the average function $\mathrm{Avg}(\cdot)$.

$$
\begin{aligned}
\frac{\|\mathrm{Avg}(X) - \mathrm{Avg}(Y)\|_2}{\|X - Y\|_2} &= \frac{|\mathrm{Avg}(X - Y)|}{\|X - Y\|_2} \\
&= \frac{|\sum_{i=1}^{h_k \cdot w_k}(X_i - Y_i)|}{\|X - Y\|_2}/(h_k \cdot w_k) \\
&\leq \frac{1}{\sqrt{h_k \cdot w_k}}.
\end{aligned}
$$

### D.4 Activation functions

Table 1 lists up the Lipschitz constants of activation functions and other nonlinear functions commonly used in deep neural networks. From Theorem 3, we only need to consider the Lipschitz constants elementwisely.

## E Lipschitz-Margin Training stabilization

We empirically found that applying the addition only when a prediction is correct stabilizes the training. Thus, in the training, we scale the addition with

$$\alpha_{F,X} := \min_{i \neq t_X}\left\{\max\left(0, \min\left(1, \frac{F(X)_{t_X} - F(X)_i}{\sqrt{2}cL_F}\right)\right)\right\}.$$

Even though $\alpha_{F,X}$ depends on $L_F$, we do not back-propagate it.

Similarly, we observed that strong regularization at initial stage of training can make training unstable. Thus, we set an initial value of $c$ small and linearly increased it to the target value in first 5 epochs as learning rate scheduling used in Goyal et al. [5].

## F Experimental setups

In this section, we describe the details of our experimental settings.

Table 2: Network structure used for experiment 6.1. For convolutional layers, output size denotes channel size of output.

|  | output size | kernel | padding | stride |
|---|---|---|---|---|
| convolution | 16 | (4,4) | (1,1) | (2,2) |
| ReLU | - | - | - | - |
| convolution | 32 | (4,4) | (1,1) | (2,2) |
| ReLU | - | - | - | - |
| fully-connected | 100 | - | - | - |
| ReLU | - | - | - | - |
| fully-connected | 10 | - | - | - |

## F.1 Experiment 6.1

### F.1.1 Base network

We used the same network, optimizer and hyperparameters with Kolter and Wong [7]. A network consisting of two convolutional and two fully-connected layers was used. Table 2 shows the details of its structure.

### F.1.2 Hyperprameters

All models were trained using Adam optimizer [6] for 20 epochs with a batch size of $50$. The learning rate of Adam was set to $0.001$. Note that these setting is the same with Kolter and Wong [7]. For a LMT model, we set $c = 1$. For an AT model, we tuned hyperparemter $c$ of C&W attack from $[0.0001, 0.001, 0.01, 0.1, 1, 10, 100, 1000, 10000]$ and chose the best one on validation data.

### F.1.3 Estimation of inequality (5)

**(A):** We calculated (A) with Proposition 2.

**(B):** We took the max of the local Lipschitz constant calculated for (C).

**(C):** First, we added a random perturbation which each element is sampled from a Gaussian with zero-mean and variance $v$ , where $v$ is set as a reciprocal number of the size of input dimension. Next, we calculated the size of a gradient with respect to the input. We repeated the above two for 100 times and used the maximum value between them as an estimation of the local Lipschitz constant.

**(D):** We used DeepFool [9].

## F.2 Experiment 6.2

Wide residual network [14] with 16 layers and a width factor $k = 4$ was used. We sampled 10000 images from an extra data available for SVHN dataset as validation data and combined the rest with the official training data, following Cisse et al. [2]. All inputs were preprocessed so that each element has a value in a range 0-1.

Models were trained with Nesterov Momentum [10] for 160 epochs with a batch size of $128$. The initial learning rate was set to $0.01$ and it was multiplied by $0.1$ at epochs $80$ and $120$. For naive models, the weight decay with $\lambda = 0.0005$ and the dropout with a dropout ratio of $0.4$ were used. For Parseval networks, the weight decay was removed except for the last fully-connected layer and Parseval regularization with $\beta = 0.0001$ was added, following Cisse et al. [2]. For a network with the spectral norm regularization, the weight decay was removed and the spectral norm regularization with $\lambda = 0.01$ was used following Yoshida and Miyato [13]. We note that both Cisse et al. [2] and Yoshida and Miyato [13] used batch-normalization for their experimental evaluations and thus, we left it for them. For LMT, we used $c = 0.01$ and did not apply weight decay. In residual blocks, the Lipschits constant for the convolutional layer and the batch normalization layer was jointly calculated as described in Sec. 5.2.

# G   Additional discussion

## G.1   Application

Since the proposed calculation method of guarded areas imposes almost no computational overhead at inference time, this property has various potential applications. First of all, we note that in real-world applications, even though true labels are not available, we can calculate the lower bounds on the size of perturbations needed to change the predictions. The primary use is balancing between the computational costs and the performance. When the provably guarded areas are sufficiently large, we can use weak and computationally cheap detectors of perturbations, because the detectors only need to find large perturbations. For data with small guarded areas, we may resort to computationally heavy options, e.g., strong detectors or denoising networks.

## G.2   Improvements from Parseval networks

Here, we discuss the difference between our work and Cisse et al. [2]. In the formulation of Parseval networks, the goal is to limit the change in some Lipschitz continuous loss by constraining the Lipschitz constant. However, since the existence of adversarial perturbations corresponds to the 0-1 loss, which is not continuous, their discussion is not applicable. For example, if we add a scaling layer to the output of a network without changing its parameters, we can control the Lipschitz constant of the network. However, this does not change its prediction and this is irrelevant to the existence of adversarial perturbations. Therefore, considering solely the Lipschitz constant can be insufficient. In LMT, the insufficiency is avoided using Proposition 1 and 2.

Additionally, we point out three differences. First, in Parseval networks, the upper bound of each component is restricted to be smaller than one. This makes their theoretical framework incompatible with some frequently used layers such as the batch normalization layer. Since they just ignore the effects of such layers, Parseval networks cannot control the Lipschitz constant of networks with normalization layers. On the other hand, our calculation method of guarded area and LMT can handle such layers without problems. Second, Parseval networks force all singular values of the weight matrices to be close to one, meaning that Parseval networks prohibit weight matrices to dump unnecessary features. As Wang et al. [12] pointed out, learning unnecessary features can be a cause of adversarial perturbations, which indicates the orthonormality condition has adverse effects that encourage the existence of adversarial perturbations. Since LMT does not penalize small singular values, LMT does not suffer the problem. Third, LMT requires only differentiable bounds of the Lipschitz constants. This lets LMT be easily extended to networks with various components. On the other hand, the framework of Parseval networks requires special optimization techniques for each component.

## G.3   Extensions of LMT

The formulation of LMT is highly flexible, so we can consider some extended versions. First, we consider the applications that require guarded area different in classes. For example, to distinguish humans from cats will be more important than to classify Angora cats from Persian cats. In LMT, such knowledge can be combined by specifying different hyperparameter $c$ for each pair of classes. Second, we consider a combination of adversarial trainings. It will be more reasonable to require smaller margins for the inputs with large perturbations. In LMT, we can incorporate this intuition by changing $c$ according to the size of perturbations or merely set $c$ to zero for perturbed data. This ability of LMT to be easily combined with other notions is one of the advantages of LMT.