[Reviews · NeurIPS 2018]

Reviewer 1



The paper presents a new strategy for training networks that are robust to adversarial attacks. The approach is intuitive and well explained: (1) The authors how that given a perturbation epsilon, the network will return the correct answer providing that the margin (difference between correct class and highest scoring incorrect class) is higher than a certain function that depends on the lipschitz constant and the norm of the pertubration. Note that a function with high margin is highly flexible/discriminative, while that with a low lipschitz constant is smooth and more robust. Based on this above, the authors introduce a regularization term (auxiliary loss) during training which encourages classifiers to be robust (i.e. increasing the "guarded area" by either increasing the margin or decreasing their lipschitz constant The authors propose a new technique for computing the Lipschitz constant that is efficient and a tighter upper bound than previous approaches. Empirical results show that the authors' work better withstands C&W style attacks compared to existing approaches.

Reviewer 2



This paper proposes a computationally efficient calculation technique that lower-bounds the size of adversarial perturbations that can deceive networks, and an effective training procedure, which robustifies networks and significantly improves the provably guarded areas around data points. The contribution of this paper is proposing an intuitive way to measure the slope to calculate the upper-bounds of gradient and provide a widely available and highly scalable method that ensures large guarded areas for a wide range of network structures. There are certain contribution and originality in the literature. Here I am concerned with the following two questions: 1. This paper defines a guarded area for a network F and a data point X as a hypersphere with a radius c. This definition is so strong that may conflict with many real world applications. 2. The authors transfer the problem into solving the Lipschitz constant and prove bounds in diffident functions and do some experiments to evaluate the performance of their method. But lack of the comparative experiments with some existing methods that make the experiment unconvincing.

Reviewer 3



This paper provides theoretical analysis for effect of perturbations on network performance. In addition it also provides way of integrating these properties with network training by using them as a additional training loss. The paper is very well written, analysis are very clear and theorems are well organized and easy to read. Good job!